# State Estimation and Remaining Useful Life Prediction of PMSTM Based on a Combination of SIR and HSMM

**Guishuang Tian [1], Shaoping Wang [1,2,3], Jian Shi [1,2,3,\*] and Yajing Qiao [1]**

1. School of Automation Science and Electrical Engineering, Beihang University, Beijing 100191, China
2. Science and Technology on Aircraft Control Laboratory, Beihang University, Beijing 100191, China
3. Ningbo Institute of Technology, Beihang University, Ningbo 315800, China
* Correspondence: shijian123@sina.com

**Abstract:** The permanent magnet synchronous traction motor (PMSTM) is the core equipment of urban rail transit. If a PMSTM fails, it will cause serious economic losses and casualties. It is essential to estimate the current health state and predict remaining useful life (RUL) for PMSTMs. Directly obtaining the internal representation of a PMSTM is known to be difficult, and PMSTMs have long service lives. In order to address these drawbacks, a combination of SIR and HSMM based state estimation and RUL prediction method is introduced with the multi-parameter fusion health index (MFHI) as the performance indicator. The proposed method's advantages over the conventional HSMM method were verified through simulation research and examples. The results show that the proposed state estimation method has small error distribution results, and the RUL prediction method can obtain accurate results. The findings of this study demonstrate that the proposed method may serve as a new and effective technique to estimate a PMSTM's health state and RUL.

**Keywords:** permanent magnet synchronous traction motor (PMSTM); state estimation; remaining useful life (RUL) prediction; hidden semi-Markov model (HSMM); sample importance resampling (SIR); multi-parameter fusion health index (MFHI)

## 1. Introduction

Urban rail transit (URT) systems have a plethora of advantages, such as large passenger volume, low pollution emission, fast operation speed and safe and punctual operation. They constitute key infrastructure components that are able to support urban economic and social development within societies [1]. URTs serve as main ways for optimizing a city's functional layout, meeting people's travel needs, alleviating urban traffic congestion and promoting economic and social development [2]. The traction motor is the core equipment of URT, and its electrical and mechanical properties are directly related to the efficiency and reliability of the whole system [3]. The permanent magnet synchronous traction motor (PMSTM) is a feasible option for URT traction motors in light of its small size, high efficiency, low moment of inertia and fast dynamic influence [4].

A PMSTM can operate for extended durations under harsh working conditions, including high speeds, large loads, strong vibrations and severe noise; and its working state is associated with the safe operation of the entire URT. If the PMSTM fails, it will cause serious economic losses and casualties. Therefore, estimating the current state of health during PMSTM operation may guide maintenance operations [5] in order to reduce downtime, increase utilization and ensure that it operates safely and efficiently under specific conditions, thereby avoiding catastrophic accidents, extending its service life and reducing property damage. The RUL of the PMSTM refers to the working time from the current moment while maintaining a certain output performance [6]. RUL prediction plays a vital role in condition-based maintenance [7], providing information critical for maintenance planning and reducing overall lifecycle costs [8]. However, it is still a challenging issue to

state estimation and RUL predictions for PMSTM due to its coupling failure mechanism, long life span and complex working environment.

Due to the importance of state estimation and RUL prediction, numerous methods have been proposed. Current methods mainly encompass both model-based and data-driven methods. Model-based methods [9] usually start from the failure mechanism of the product and obtain the physical model of performance degradation via construction of a relationship between performance degradation amount and stress (high temperature, high pressure, strong vibration, etc.) in order to analyze the health state and RUL of the system [10]. In other words, mathematical representations of interactions between processes and components must be defined, and precise solutions can only be achieved when precise failure behavior degradations are available [11]. Any PMSTM has strong nonlinearity and coupling; its internal mechanism is complex, and its environment and working conditions are diverse. Thus, it is very difficult to establish a complete physical model of the performance degradation of a PMSTM. In contrast, the use of data-driven methods has gained traction due to the lack of physical knowledge and the continuous advancement of modern sensor systems and data storage/analysis techniques [12], which are more suitable for PMSTMs.

Current mainstream data-driven methods include machine learning and mathematical statistics. Machine learning methods used for state estimation and RUL prediction mainly include neural networks and support vector machines (SVMs). The application of neural networks in the field of state estimation and life prediction is continually expanding, which has since become one of the important branches of prediction algorithms. Shifat and Zhang et al. [13,14] proposed RUL prediction methods based on the recurrent neural network (RNN). Li, Sateesh and Chen et al. [15–17] proposed methods for RUL estimation using deep convolutional neural networks (DCNNs). Gougam et al. [18] described a neuro-fuzzy based method for RUL prediction. Moreover, Kewalramani et al. [19] utilized a feedforward neural network to estimate the RUL of the motor. However, machine learning requires a large amount of training data, which is not suitable for research objects with few datasets, such as PMSTM. SVM has obvious advantages in solving small sample and nonlinear problems of life prediction. In Reference [20], a novel partial discharge data (PDD)-based SVM model was proposed for the RUL prediction of batteries. García et al. [21] utilized the particle swarm optimization (PSO) algorithm to optimize the SVM kernel function. Compared to neural networks, SVM requires fewer learning samples, can be easily converged to the global optimal solution and has strong generalization ability. However, selecting kernel functions and model parameters in SVM is difficult, thereby hampering the improvement of algorithm performance.

Statistical methods include the Wiener process, Gaussian model, hidden Markov model (HMM), Kalman filter and particle filter. Le et al. [22] combined the Wiener process with principal component analysis (PCA) to propose a probabilistic method for RUL prediction. The Wiener process has gained a number of achievements in regard to life prediction of linear degradation [23]; however, its ability for nonlinear expression requires further study. Liu et al. [8] proposed a GMM-based RUL estimation method by exploiting the ability of the Gaussian mixture model (GMM) in terms of learning complex joint probability density functions from data. HMMs are used to characterize discontinuous performance degradation processes in life prediction. Gao [24] divided the whole RUL prediction process into three parts, including offline modeling, online state estimating and online life predicting. In the offline modeling part, HMM and proportional hazard model (PHM) were built to map the whole degradation path. However, the time-invariant geometric distribution of one-step transition probabilities and durations resulted in HMMs being unable to match the real degradation process well. Hidden semi-Markov models (HSMMs) incorporate self-transition probabilities into the distribution of state durations and are widely used in health monitoring [25]. In reference [26–28], the dependence of the duration of adjacent degraded states in the HSMM has been described and modeled, resulting in more efficient and accurate online estimation of degraded states and the distribution

of RUL. References [29–31] used different styles of Kalman filters to estimate the RULs of devices, such as motors and bearings. Jouin et al. [32] described a physics-based model that utilized particle filters, representing the posterior distribution as the number of particles. The different combinations used in previous works included data size, allowable noise and bias in data, actual loading conditions and the complexity of degradation that identifies the deteriorated behavior for predicting future health. In reference [33], a model-based method for predicting the RUL of PMSMs using phase current and vibration signals was proposed. The proposed method included feature selection and RUL prediction based on a particle filter with a degradation model. Unfortunately, Kalman and particle filters always require assigning an analytical form to state and observation equations [25], making both methods less convenient than the HSMM. However, the HSMM must return from the current moment to the initial moment, whereas the PMSTM has a long lifespan, thereby making RUL estimation both cumbersome and computationally expensive. The sample importance resampling (SIR) filter algorithm, a type of particle filter, is able to solve this problem and directly recurse the observation sequence of the hidden state from the previous state. Therefore, this paper combines SIR and the HSMM to propose a PMSTM's performance state recognition and RUL prediction method, and replaces the backtracking process when the traditional HSMM takes the optimal hidden state sequence by recurrence estimation.

In a data-driven approach, state estimation and RUL prediction correspond to multi-source condition monitoring data. Therefore, it is important to extract key parameters characterizing the system degradation state from monitoring data [6]. In order to monitor the state of the system, numerous sensors must be adopted to collect data. Since the data are multi-dimensional and different sensor data show different patterns, in order to better evaluate the degradation of the system and estimate the RUL, the dimensionality of the data should be reduced [34] to generate a health index (HI), which may be achieved in various ways. For example, Hu et al. [35] used isometric mapping (ISOMAP) as a feature reduction method to generate one-dimensional HI. However, this method loses too much information during the feature reduction process. Liu et al. [36] proposed a composite HI by weighted summation based on multiple degraded sensor data, in which the weights are determined by optimizing a quadratic programming problem. Yang et al. [11] proposed a general HI dynamic smoothing algorithm and implemented the prediction framework with exponential HI degradation as an example. Ahmad et al. [37] inferred a bearing's health through a dimensionless HI. The HI measures the instantaneous vibration level of the bearing with respect to a normal baseline value. By the amount of redundant information between multiple sets of feature parameters, the difficulty of building a performance degradation prediction model is greatly increased. In addition, the PMSTM's feature parameters are nonlinear, making it necessary to investigate a new form of HI construction. Hence, we consider the objective weight evaluation of each feature index and reduce the correlation, thereby integrating the MFHI.

Therefore, this paper uses the multi-parameter fusion health index (MFHI) as an indicator and combines the HSMM with SIR to propose a state estimation and RUL prediction method for PMSTM. The flow chart is shown in Figure 1, including four parts: state monitoring data preprocessing and MFHI construction; observation sequence acquisition; HSMM training; current health state recursive estimation and RUL prediction. The main research points are summarized as follows. In order to realize the prioritization of the state-characteristic parameters, this study adopts the entropy weight method to objectively determine the weights of multiple indicators. Moreover, PCA is used to reduce the correlation between the selected performance parameters. Then, according to the corresponding weight assignments, fusion is performed to construct a MFHI that represents the health state of PMSTM. After the MFHI is low-pass filtered, the state is then divided to obtain the observation sequence. HSMM is subsequently used to describe the relationship between the internal state and the observation sequence of PMSTM, after which the overall performance degradation model is established. Furthermore, the observation sequence of the hidden

state is directly recursive from the previous state through SIR to attain the recursive optimal estimation of the health state and prediction of RUL.

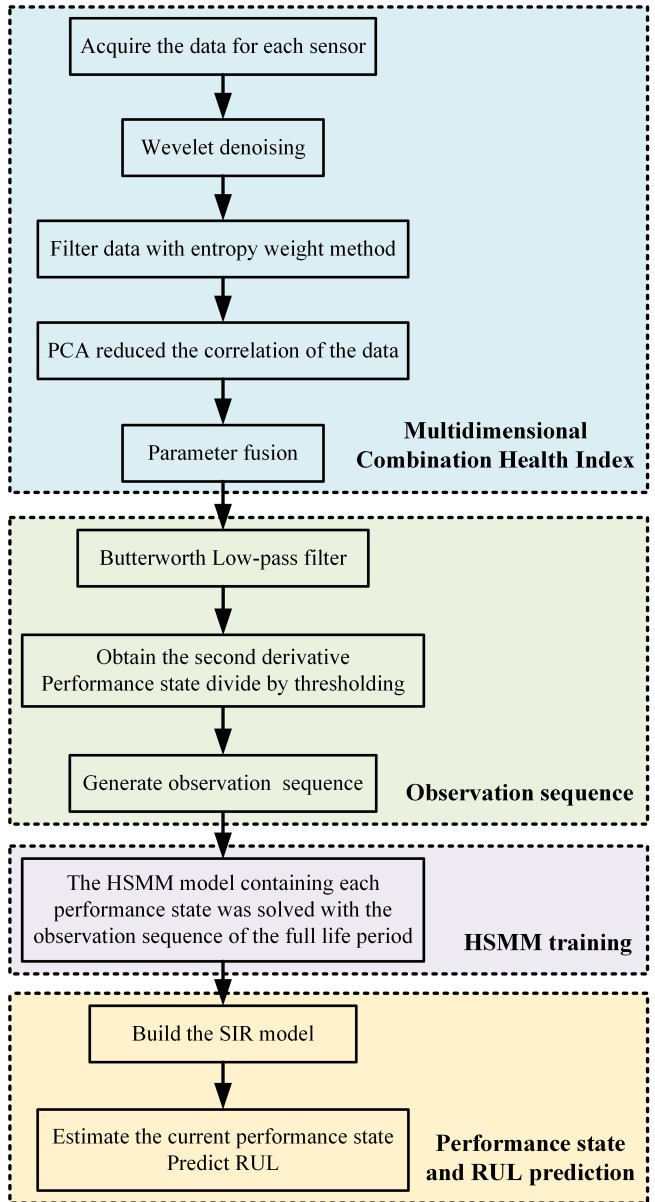

**Figure 1.** The flow diagram of state estimation and RUL prediction.

The remainder of this paper is organized according to the following outline. The MFHI is constructed in Section 2. Section 3 clarifies the state estimation and RUL prediction methods for combining SIR with HSMM. Section 4 summarizes the proposed method. Section 5 analyzes the proposed method's results with a corresponding discussion. Finally, Section 6 provides the conclusions and future work ascertained from the findings of this study.

## 2. MFHI Construction

It is vital to find suitable feature parameters for state estimation and RUL prediction of an PMSTM. Multiple sets of performance parameters can more comprehensively characterize the health state. Therefore, the entropy weight method is used in this study to determine the weight of multiple indexes, and the correlation between the selected performance parameters is then reduced by PCA. Finally, the corresponding weight is assigned and fused to construct the MFHI.

### 2.1. Wavelet Denoising

PMSTMs operate in complex environments and working conditions. It is inevitable to incur noise during data acquisition. Excessive noise will affect the accuracy of the subsequent feature extraction analysis results. Therefore, noise reduction processing should be initially performed for the collected sensor signals. The wavelet denoising method is adopted in this paper. The original condition monitoring data are a signal sequence $S(t)$ whose length is $G$, which is superimposed with noise:

$$S(t) = z(t) + e(t) \tag{1}$$

where $z(t)$ is the actual signal, $e(t)$ is the noise and $t$ is the sampling interval point.

Orthogonal wavelet decomposition using the Mallat algorithm:

$$\begin{cases} S_{p,k} = \sum_G S_{p-1,G} \cdot l_{G-2k} \\ D_{p,k} = \sum_G S_{p-1,G} \cdot o_{G-2k} \end{cases} \tag{2}$$

where $S_{p,k}$ represents the scale coefficient, $D_{p,k}$ represents the wavelet coefficient, $l_{G-2k}$ and $o_{G-2k}$ are a pair of orthogonal mirror filter banks and $p$ is the number of decomposition layers.

Here, $S$ is decomposed into different frequency bands, and a fixed threshold is then selected to process the noisy signal at each layer. Inverse wavelet transformation is then performed on the denoised wavelet coefficients to obtain the reconstructed signal $S^{(1)}$:

$$S^{(1)} = \sum_G S_{p,G} \cdot l_{G-2k} + \sum_G D_{p,G} \cdot o_{G-2k} \tag{3}$$

### 2.2. Filter Data with Entropy Weight Method

Assuming that the evaluation index of the target consists of $\phi$, namely, $S^{(1)} = (S_1^{(1)}, S_2^{(1)}, \cdots, S_\phi^{(1)})$, the decision matrix is

$$S^{(1)} = \begin{bmatrix} S_{11}^{(1)} & \cdots & S_{1\phi}^{(1)} \\ \vdots & \ddots & \vdots \\ S_{G1}^{(1)} & \cdots & S_{G\phi}^{(1)} \end{bmatrix} \tag{4}$$

$S_{gu}^{(1)}(g = 1, 2, \cdots, G; u = 1, 2, \cdots, \phi)$ in the matrix is the $g$th value of the index $u$.

In order to eliminate the issue in which the final results are unable to be compared due to different index quantities, the initial decision matrix must be dimensionless—that is, normalized data processing. A Z-score-standardized decision matrix $S^{(2)} = (S_{gu}^{(2)})_{G \times \phi}$ is then constructed.

Under the definition of $\phi$ evaluation indicators, the entropy values of the evaluation indicators can be then determined:

$$H_u = \frac{\sum_{g=1}^{G} f_{gu} \ln(f_{gu})}{- \ln(G)}$$

$$f_{gu} = S_{gu}^{(2)} / \sum_{g=1}^{G} S_{gu}^{(2)} \tag{5}$$

The difference between the entropy value of the index and one is used as the degree of information availability of the index. A larger entropy indicates less information availability, and vice versa. The entropy weight of each indicator is finally obtained as:

$$\omega_u = \frac{1 - H_u}{\phi - \sum_{u=1}^{\phi} H_u}, \quad \sum_{u=1}^{\phi} \omega_u = 1 \tag{6}$$

$S^{(2)} = (S_1^{(2)}, S_2^{(2)}, ..., S_\phi^{(2)})$ is filtered to $\sigma$ performance parameters according to the weights, which is reordered to get $S^{(3)} = (S_1^{(3)}, S_2^{(3)}, \cdots, S_\sigma^{(3)})$.

### 2.3. PCA Reduces the Correlation of the Data

Through the entropy weight method, the parameters most related to the performance degradation of PMSTM can be screened out from many monitoring parameters, but a certain level of correlation may exist between the parameters. Therefore, reducing the correlation between the filtered performance parameters by PCA is necessary. The specific steps of the PCA algorithm pertaining to the extraction of principal components of the screened $\sigma$-dimensional monitoring parameters are as follows:

1.  Calculate the covariance matrix $COV$ of $S^{(3)}$

$$COV = \frac{1}{\sigma - 1} \left( S^{(3)} \right)^T \cdot S^{(3)} \tag{7}$$

2.  Calculate the eigenvalues of $COV$, sorting from largest to smallest to get $\varphi_1, \varphi_2, \cdots, \varphi_\sigma$, and obtain the corresponding eigenvector $\kappa_1, \kappa_2, \cdots, \kappa_\sigma$.
3.  Obtain the principal components, where $s = 1, 2, \cdots, \sigma$

$$K_s = S^{(3)} \cdot \kappa_s \tag{8}$$

4.  Sort the principal components to get the cumulative contribution rate $CON$:

$$CON(s) = \sum_{\iota=1}^{s} \varphi_\iota / \sum_{\iota=1}^{\sigma} \varphi_\iota \tag{9}$$

When the cumulative contribution rate exceeds a certain value, the corresponding first $\zeta$ principal components should be adopted, where $\zeta$ is less than the total number of principal components and $\zeta < \sigma$, thereby obtaining $S^{(4)} = (S_1^{(4)}, S_2^{(4)}, \cdots, S_\zeta^{(4)})$.

### 2.4. Parameters Fusion

In order to reduce the influence of the performance parameter range, it is necessary to normalize the performance parameters through dispersion standardization while fusing multiple performance parameters into a single performance index via parameter fusion. The index is MFHI, $HI = (HI_1, HI_2, \cdots, HI_G)$, and has the following formula:

$$HI^0 = S^{(4)} \sum_{z=1}^{\zeta} \rho_z$$

$$\rho_z = \varphi_z / \sum_{z=1}^{\zeta} \varphi_z \tag{10}$$

$$HI = \frac{HI^0 - HI_{\min}^0}{HI_{\max}^0 - HI_{\min}^0}$$

where $\rho_z$ is the weight value of $\zeta$ performance parameters after dimensionality reduction; $HI_{\min}^0$ and $HI_{\max}^0$ are the maximum and minimum values in the data, respectively.

## 3. State Estimation and RUL Prediction Combining SIR and HSMM

### 3.1. Observation Sequence Acquisition

The description of the performance of a component is usually divided into its normal state, mild degradation state, moderate degradation state and severe degradation state. After a device reaches a state of severe degradation, it will fail over a period of

time. Accordingly, it is represented by dividing the MFHI into five categories, denoted as $\{\{HI\}_c\}, c = 1, 2, \cdots, 5$.

$HI(t)$ is smoothed with a Butterworth low-pass filter, for which the amplitude square function and performance index are:

$$|J_a(j\Omega)|^2 = \frac{1}{1 + \left(\frac{\Omega}{\Omega_c}\right)^{2C}}$$

$$A_p = 10 \lg\left(1 + \varepsilon_p^2\right), A_s = 10 \lg\left(1 + \varepsilon_s^2\right) \tag{11}$$

where $C$ is the order of the filter, $\Omega$ is the frequency domain center, $\Omega_c$ is the cutoff frequency, $\varepsilon_p$ is the passband attenuation and $\varepsilon_s$ is the stopband attenuation.

The transfer function can be expressed as

$$J(t) = \frac{\sum_{f=0}^{E} v_f HI(t - f)}{\sum_{f=1}^{F} c_f HI_F(t - f)} \tag{12}$$

Then, the filtering result is

$$HI_F(t) = \sum_{f=0}^{\infty} J(f) HI(t - f) = -\sum_{f=1}^{F} c_f HI_F(t - f) + \sum_{f=0}^{E} v_f HI(t - f) \tag{13}$$

The second derivative is found and divided into five states according to its threshold. $HI_F(t)$ of different stages and their corresponding states are labeled (Table 1) to obtain the observation sequence $Y(t) = \{y_1, y_2, \cdots, y_G\}$.

**Table 1.** State label.

| State | Normal | Mild Degradation | Moderate Degradation | Severe Degradation | Failure |
|-------|--------|------------------|----------------------|--------------------|---------|
| Label | 1 | 2 | 3 | 4 | 5 |

*3.2. HSMM Training*

The observation sequence of the full life period is used to solve HSMM $\lambda = (\mathbf{\Pi}, A_0, B, \Theta)$, in which the definitions of the parameters are:

1. The initial state probability distribution:

$$\mathbf{\Pi} = \{\pi_i\}, \pi_i = P(x_0 = c), 1 \leq c \leq 5 \tag{14}$$

where $x_0$ is the health state of the PMSTM at the initial moment.

2. The state transition probability matrix, which represents the probability of transition between states during the operation of PMSTM:

$$A_0 = \{a_{ij}\}, \quad a_{ij} = P(x_{t+1} = j \mid x_t = i), 1 \leq i < j \leq 5 \tag{15}$$

where $a_{ij}$ represents the probability that the PMSTM transitions from state $i$ to state $j$ during the running process.

3. The observed state probability matrix:

$$B = \{b_{cq}\}, \quad b_{cq} = P(y_q \mid x_t = c), 1 \leq q \leq Q \tag{16}$$

where $Q$ is the number of observable states of PMSTM, and $b_{cq}$ represents the probability of observing the $q$th observable state when the health state is $x_t = c$.

4. The dwell time distribution for each state:

$$\Theta = \{\theta_c\} \tag{17}$$

where $\theta_c = \left(\mu_{x_c}, \delta_t^2(c)\right)$ is the parameter of the probability density function. $\mu_{d_c}$ is the mean value and $\delta_t(c)$ represents the proportion of different states.

Assuming that the likelihood function of HSMM is $P(Y \mid \lambda)$, HSMM training involves obtaining the parameter that maximizes $P(Y \mid \lambda)$ from the observation sequence sample $Y$. By employing the Baum–Welch algorithm, the parameters of the model can be solved. The training process is described below.

1. Through the current model parameters $\bar{\lambda}$, the expectation of $P(Y, X \mid \lambda)$ under condition $P(X \mid Y, \bar{\lambda})$ is obtained by combining the Viterbi algorithm, the forward algorithm and the backward algorithm.

   - According to the current observation sequence, the most likely hidden state sequence is obtained by the Viterbi algorithm.
     Calculate the local state at the initial moment:

$$\delta_1(i) = \pi_i b_i(y_1)$$
$$\Psi_1(i) = 0 \tag{18}$$

   and recurse

$$\delta_t(j) = \max_{1 \le i \le 5} \left[\delta_{t-1}(i) a_{ij}(x_{t-1})\right] b_j(y_t)$$
$$\Psi_t(j) = \arg \max_{1 \le i \le 5} \left[\delta_{t-1}(i) a_{ij}(x_{t-1})\right]. \tag{19}$$

   The maximum $\delta_t(i)$ at time $t$ is the probability of the most likely hidden state. The auxiliary variable $\Psi_t(j)$ is used to store the optimal state of PMSTM at time $t - 1$ under the condition that time $t$ is in state $j$. Thus:

$$x_t^* = \arg \max_{1 \le j \le 5} [\delta_t(j)] \tag{20}$$

   Backtracking to get the sequence of hidden states:

$$x_{t-1}^* = \Psi_t(x_t^*) \tag{21}$$

   so the dwell time can be obtained:

$$d_t(j) = x_t^*(j) \cdot x_{t-1}^*(j) \cdot d_{t-1}(j) + 1 \tag{22}$$

   - Calculate variables using forward and backward algorithms.
     Calculate the forward probability of each hidden state at the initial moment:

$$\alpha_1(i) = \pi_i b_i(y_1)$$
$$A_{x_1} = P(x_1) + (X - P(x_1)) \cdot A_0 \tag{23}$$

   get forward variable $\alpha_{t+1}(i)$

$$\alpha_{t+1}(j) = \left[\sum_{i=1}^{5} \alpha_t(i) a_{ij}(x_t)\right] b_j(y_{t+1})$$
$$A_{x_{t+1}} = P(x_{t+1}) + (X - P(x_{t+1})) \cdot A_0 \tag{24}$$

   Calculate variables $\beta_t(i)$ using the backward algorithm

$$\beta_T(i) = 1$$
$$\beta_t(i) = \left[\sum_{i=1}^{5} a_{ij}(x_t) b_j(y_{t+1})\right] \beta_{t+1}(j) \tag{25}$$

and get the expectation of $P(Y, X \mid \bar{\lambda})$ under the condition of $P(X \mid Y, \bar{\lambda})$

$$L(\lambda, \bar{\lambda}) = \sum_X P(X \mid Y, \bar{\lambda}) \log P(Y, X \mid \lambda) \tag{26}$$

2. The parameters of the model can be updated by maximizing the expected value

$$\bar{\lambda} = \arg\max_\lambda \sum_X P(X \mid Y, \bar{\lambda}) \log P(Y, X \mid \lambda) \tag{27}$$

The equation for updating the model parameters can then be obtained:

$$\bar{\pi}_i = \gamma_1(i) \tag{28}$$

$$\bar{a}_{ij}^0 = \frac{\left(\sum_{t=1}^{T-1} \xi_t(i,j)\right) \odot G}{\left(\sum_{t=1}^{T-1} \gamma_t(i)\right) \odot G} \tag{29}$$

$$\xi_t(i,j) = P(x_t = i, x_{t+1} = j \mid Y, \lambda) = \frac{\alpha_t(i) a_{ij}(x_t) b_j(y_{t+1}) \beta_{t+1}(j)}{\sum_{i=1}^{5} \sum_{j=1}^{5} \alpha_t(i) a_{ij}(x_t) b_j(y_{t+1}) \beta_{t+1}(j)} \tag{30}$$

$$\gamma_t(i) = \frac{\alpha_t(i)}{\sum_5 \alpha_t(i)} \tag{31}$$

$$\mu_{i,d} = \frac{\sum_{t=1}^{T-1} \alpha_{t=1}(i) \left(\sum_{j=1, j \neq i}^{5} a_{ij}(x_t(i)) b_j(y_{t+1}) \beta_{t+1}(j)\right) d_t(i)}{\sum_{t=1}^{T-1} \alpha_{t=1}(i) \left(\sum_{j=1, j \neq i}^{5} a_{ij}(x_t(i)) b_j(y_{t+1}) \beta_{t+1}(j)\right)} \tag{32}$$

$$\delta_{i,d}^2 = \frac{\sum_{t=1}^{T-1} \alpha_{t=1}(i) \left(\sum_{j=1, j \neq i}^{5} a_{ij}(d_t(i)) b_j(y_{t+1}) \beta_{t+1}(j)\right) (d_t(i) - \mu_{i,d})^2}{\sum_{t=1}^{T-1} \alpha_{t=1}(i) \left(\sum_{j=1, j \neq i}^{5} a_{ij}(d_t(i)) b_j(y_{t+1}) \beta_{t+1}(j)\right)} \tag{33}$$

$$b_j(l) = \frac{\sum_{t=1, y_t=Y_l}^{T} \gamma_t(j)}{\sum_{t=1}^{T} \gamma_t(j)} \tag{34}$$

Thus, the parameters of the HSMM can be determined. The iterative update procedure of the model parameters is summarized in Algorithm 1.

---

**Algorithm 1** HSMM training procedure.

---

1. **Input** Observation sequence $Y(t) = \{y_1, y_2, \cdots, y_G\}$ and initial model parameters $\bar{\lambda}$.
2. **Calculate** the sequence of hidden states and the dwell time based on Equations (21) and (22).
3. **Calculate** forward variable $\alpha_{t+1}(i)$ and backward variable $\beta_t(i)$ using the forward and backward algorithms, see Equations (23)–(25).
4. **Calculate** the expectation of $P(Y, X \mid \bar{\lambda})$ under the condition of $P(X \mid Y, \bar{\lambda})$ based on Equation (26).
5. **Update** the model parameters by maximizing the expectation value based on Equation (27).
6. **Determine** whether the model parameters converge, if not, turn to step 2, and if they converge, the equation for updating the model parameters is obtained to determine the model parameters.

---

### 3.3. Recurrent Estimation of Current Health State

According to the MFHI and observation sequence $Y(t) = \{y_1, y_2, \cdots, y_G\}$ of the PMSTM, the observation sequence of the hidden state can be directly found by recursion from the previous state through SIR, and the recursive optimal estimation of the health

state of the PMSTM is then realized. The state estimation process is shown in Figure 2. The specific steps taken are as follows.

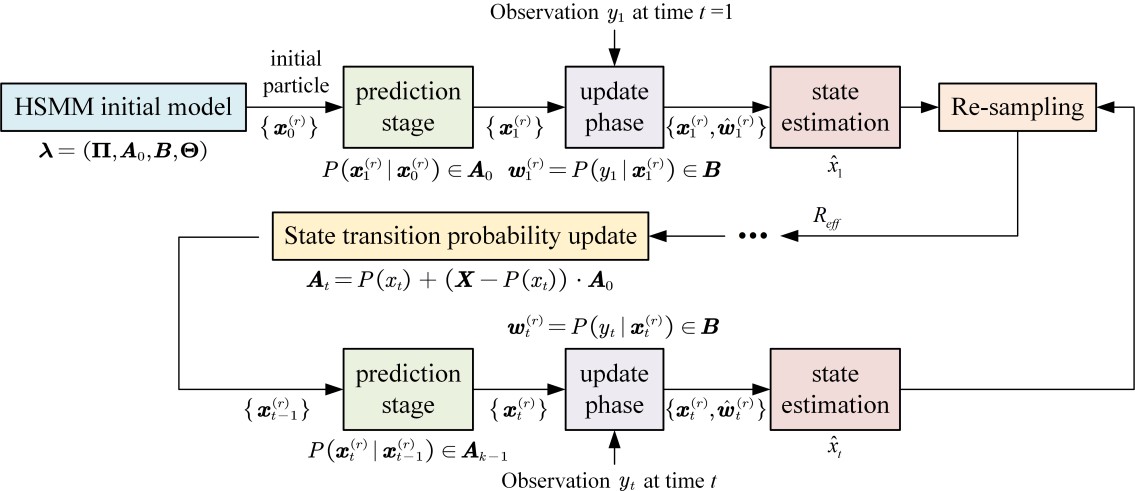

**Figure 2.** The process of recursive estimation of health status.

1. Generate an initial particle set $\left\{x_0^{(r)}\right\}, 1 \leq r \leq R$ according to the state probability distribution $\Pi$ at the initial moment.

2. State transition (prediction): According to the particle set $\left\{x_{t-1}^{(r)}\right\}$ obtained at time $t-1$, the particle set $\left\{x_t^{(r)}\right\}$ of the state at time $t$ is obtained through the state transition probability matrix $A_{t-1}$:

$$x_t^{(r)} : P\left(x_t^{(r)} \mid x_{t-1}^{(r)}\right) \in A_{t-1} \tag{35}$$

3. Calculate particle weights (update): According to the observed value $y_t$ at time $t$ and the observed state probability matrix $B$, the weight value $w_t^{(r)}$ of each predicted particle is obtained:

$$w_t^{(r)} = P\left(y_t \mid x_t^{(r)}\right) \in B \tag{36}$$

4. Normalize the calculated weight value of each particle:

$$\hat{w}_t^{(r)} = \frac{w_t^{(r)}}{\sum_{r=1}^{R} w_t^{(r)}} \tag{37}$$

5. State estimation: Calculate the estimated value of the current health state according to the particle set at time $t$ and the weight, $\left\{x_t^{(r)}, \hat{w}_t^{(r)}\right\}$, of each particle:

$$\hat{x}_t = \sum_{r=1}^{R} \hat{w}_t^{(r)} \cdot x_t^{(r)} \tag{38}$$

6. Resampling: Calculate the number of effective particles according to the normalized weight $\hat{w}_t^{(r)}$ of each particle, and resample and update the particle set as the particle set for state estimation at the next moment. The effective particle number $R_{eff}$ can be calculated as:

$$R_{eff} = \frac{R}{\sum_{r=1}^{R} \hat{w}_t^{(r)}} \tag{39}$$

7.  State transition probability matrix update: Calculate a new state transition probability matrix $A_t$ according to the residence time $d_t$ of each healthy state of the PMSTM:

$$A_t = P(x_t) + (X - P(x_t)) \cdot A_0 \qquad (40)$$

where

$$d_t = \hat{x}_t \cdot \hat{x}_{t-1} \cdot d_{t-1} + 1 \qquad (41)$$

By following the above process, recursive estimation of the current health state of the PMSTM can be achieved.

### 3.4. RUL Prediction

The RUL of the PMSTM is determined according to the remaining time of the current health state and the dwell time of each health state after the current health state. After determining the current health state $\hat{x}_t$ of the PMSTM, the RUL calculation can be divided into the following two parts:

1.  Calculate the remaining time of the current state.

$$d(x_t) = \sum_{i=1}^{5} \left( \mu_{d_i} - d_t(i) \right) \odot \delta_t(i) \qquad (42)$$

An estimate of the remaining time of the PMSTM in this state can be obtained by weighted summation.

2.  Calculate the remaining time of the subsequent state.
    Calculate the next state according to the initial state transition probability matrix $A_0$ until the failure state. The probability that the next state of the PMSTM may appear is defined as:

$$\delta_{\text{next}} = \left[ \delta_{t+\tilde{d}}(i) \right]_{1 \leq i \leq 5} = (A_0)^T \cdot \delta_t(i) \qquad (43)$$

The highest probability is the state that may appear at the next moment:

$$x_{next} = x_{t+\tilde{d}} = \arg\max_{1 \leq i \leq 5} \delta_{t+\tilde{d}}(i) \qquad (44)$$

If $x_{t+\tilde{d}}$ reaches a failure state, the PMSTM will fail when the dwell time is reached in that state. Calculate the remaining time in each state:

$$d\left(x_{t+\tilde{d}}\right) = \sum_{i=1}^{5} \mu_{d_i} \odot \delta_{t+\tilde{d}}(i) \qquad (45)$$

Iteratively calculate this in view of the above process, after which the RUL prediction value of PMSTM can be obtained:

$$RUL = \sum d \qquad (46)$$

## 4. Proposed Method

In this study, we propose a fusion method of HI and a combination of SIR and HSMM to estimate the state and predict the RUL of PMSTM. Figure 3 illustrates the proposed method. The sensor signal $S(t)$ of the PMSTM is denoised by wavelet transformation to obtain $S^{(1)}(t)$. After Z-score normalization of the noise reduction data, we can get $S^{(2)}(t)$. The entropy weight method is used to determine the weights of multiple indicators and filter the indicators $S^{(3)}(t)$ with larger weights. The correlations between the selected indicators are then reduced by PCA to obtain $S^{(4)}(t)$, and the corresponding weight is assigned and fused to obtain the MFHI $HI(t)$. The MFHI is subjected to low-pass filtering and state partitioning to obtain the observation sequence $Y(t)$. HSMM training is then performed to establish an overall performance degradation model $\lambda = (\Pi, A_0, B, \Theta)$.

Further, the PMSTM state estimation and RUL prediction are achieved through recursion of the sequence of observations corresponding to the hidden state by SIR.

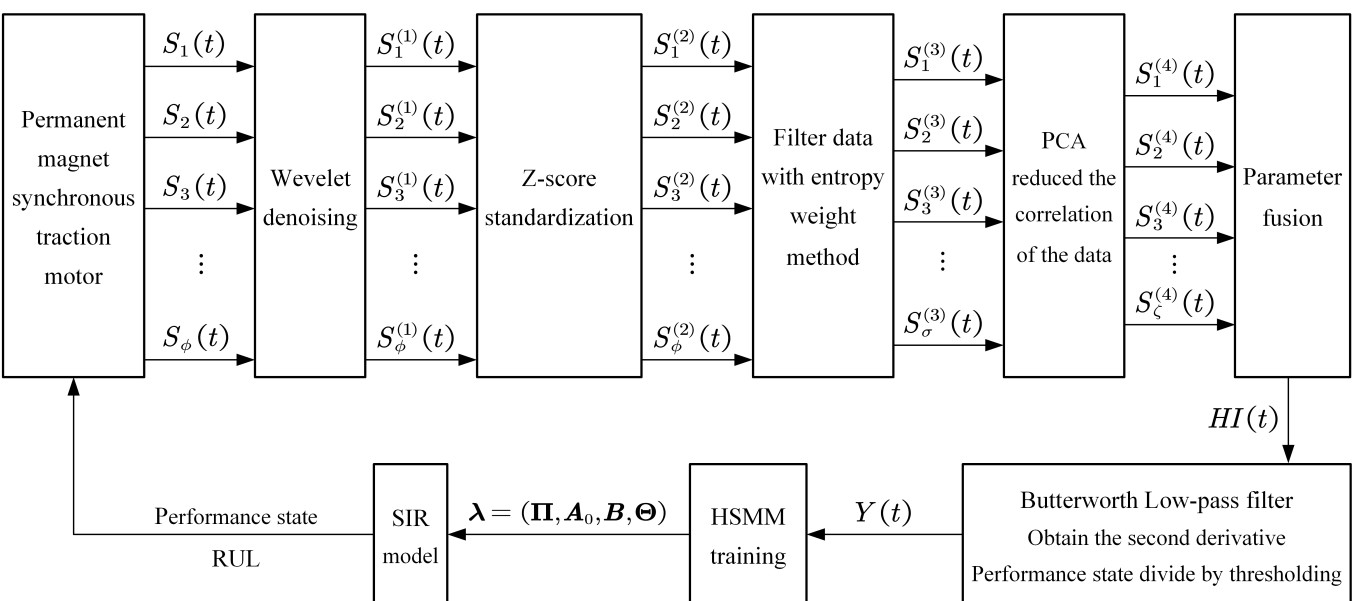

**Figure 3.** State estimation and RUL prediction method for PMSTMs based on a combination of SIR and HSMM.

## 5. Experimental Details and Analysis of Results

The original signal represents the full-life data that were collected from eight sensors (current, voltage, two temperatures, two vibrations, torque and rotational speed) during PMSTM operation. In order to validate the proposed state estimation and RUL prediction method, 70% of the MFHI extracted from the state monitoring data were used as training samples, and the remaining 30% were used as test samples.

Experiments were carried out in MATLAB. Db4 was used as the wavelet base function with a hierarchy of three; and the hard threshold, soft threshold and fixed threshold were selected so as to denoise the original signal. The noise reduction results of the rotational speed signal of PMSTM were shown in Figure 4.

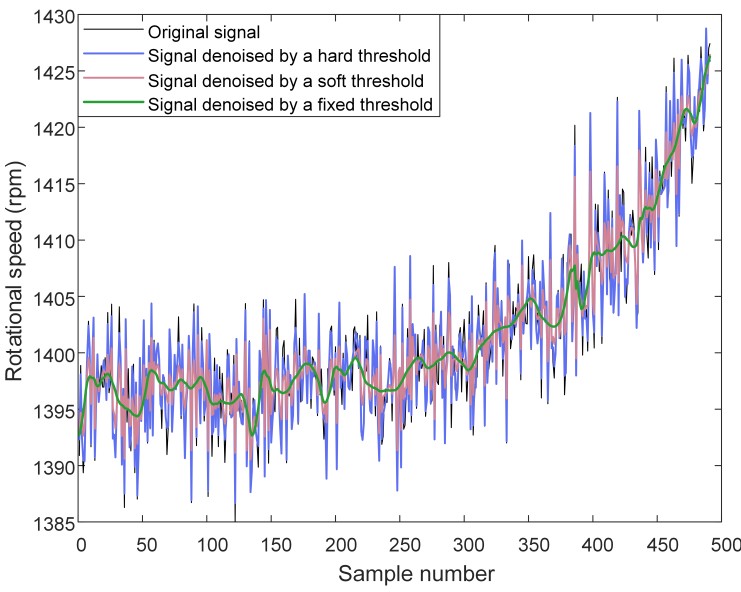

**Figure 4.** Before and after noise reduction of signal.

The signal-to-noise ratio (SNR) and root mean square error (RMSE) after noise reduction for all signals are shown in Table 2. The use of wavelet transformation to eliminate noise of the signal is shown to greatly improve SNR and reduce RMSE. The RMSE are quite different due to the magnitude and units of the different signals being different. The SNR and RMSE treated with the three threshold treatments are not much different, but it was more advantageous to choose the fixed threshold for filtering. Moreover, combined with Figure 4 for comparison, it was decided to choose the fixed threshold for signal noise reduction.

**Table 2.** SNR and RMSE for signal denoising.

| Signal Number | The Hard Threshold | | The Soft Threshold | | The Fixed Threshold | |
|---|---|---|---|---|---|---|
| | SNR (dB) | RMSE | SNR (dB) | RMSE | SNR (dB) | RMSE |
| 1 | 22.6290 | 1.7373 | 22.6290 | 1.7373 | 22.6290 | 1.7373 |
| 2 | 17.7301 | 72.3660 | 17.7301 | 72.3660 | 17.9301 | 72.3662 |
| 3 | 18.7183 | 5.4896 | 18.7183 | 5.4896 | 18.7183 | 5.4899 |
| 4 | 28.8169 | 295.2581 | 28.9579 | 290.4806 | 28.9879 | 289.4699 |
| 5 | 58.7933 | 2.7439 | 58.7933 | 2.7439 | 58.7933 | 2.7437 |
| 6 | 17.5773 | 69.3978 | 17.5773 | 69.3978 | 17.5773 | 69.3973 |
| 7 | 21.9337 | 3.1351 | 21.9337 | 3.1351 | 21.9335 | 3.1351 |
| 8 | 17.7781 | 181.0096 | 18.5190 | 166.2539 | 18.8581 | 159.9033 |

The entropy weight method was used to prioritize the state feature parameters after signal denoising, as shown in Figure 5a. Moreover, the top five parameters closely related to the performance degradation of PMSTM were screened out.

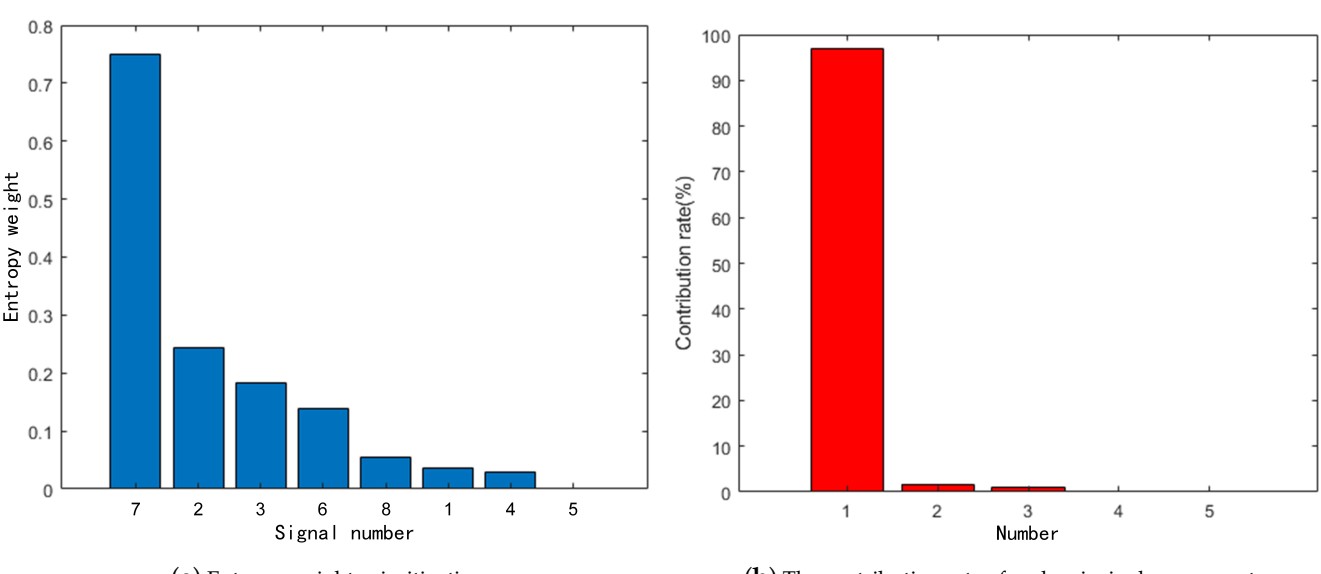

(**a**) Entropy weight prioritization　　　　(**b**) The contribution rate of each principal component

**Figure 5.** Data filtering.

PCA transformation was then performed on the reserved performance parameters, of which the contribution rate of each principal component is shown in Figure 5b. Here, the contribution rate of the first principal component was shown to reach 96.5%, which was much higher than the contribution rates of the other principal components. However, in order to reduce the data's dimensions to the greatest extent, the data characteristics of the original data should be retained as much as possible. Accordingly, the cumulative contribution rate was set to 97%, and the first two principal components were kept.

The MFHI that was obtained following parameter fusion is shown in Figure 6a. According to the division of the number of internal health states and the description of the

transition process between the states (Figure 6), the observed state was divided into five states.

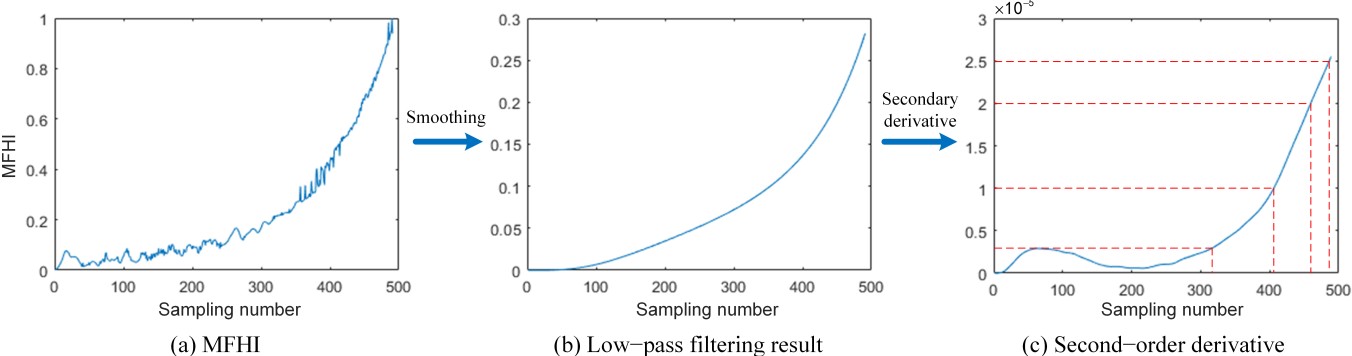

(a) MFHI      (b) Low−pass filtering result      (c) Second−order derivative

**Figure 6.** The process of performance-status classification.

The performance of the system's output is related to the internal state, where the internal state determines the observed state. Each theoretical observed state corresponded to a hidden state, as shown in Figure 7.

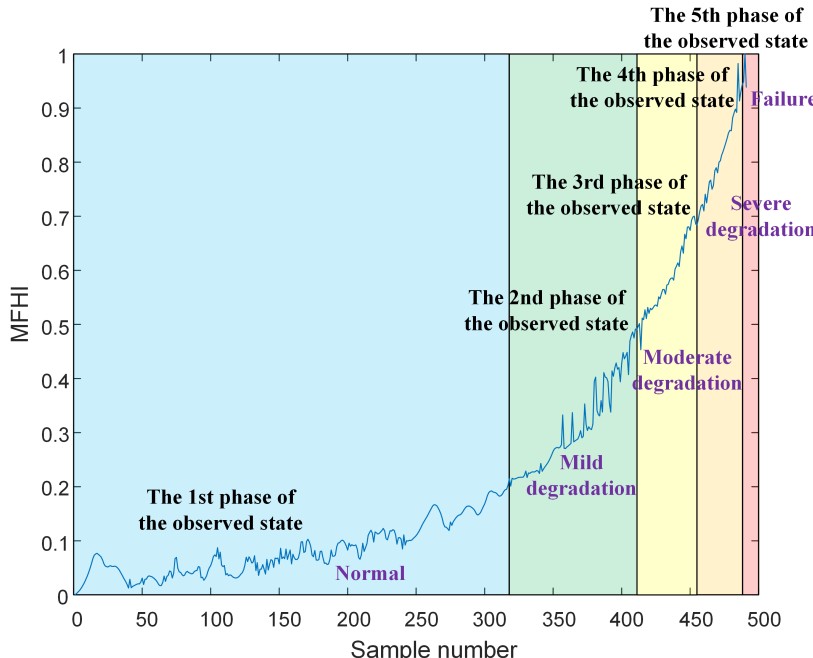

**Figure 7.** The division of health state and observation state.

The MFHIs were labeled at different stages with their corresponding states. The observation sequence that was obtained after labeling is shown in Figure 8.

HSMM training was then performed. The model parameters were initialized, and they were solved iteratively according to the model parameter solution. The performance degradation model parameters of PMSTM were obtained as

$$\mathbf{\Pi} = [1, 0, 0, 0, 0] \tag{47}$$

$$A^0 = \begin{bmatrix} 0 & 1 & 0 & 0 & 0 \\ 0 & 0 & 1 & 0 & 0 \\ 0 & 0 & 0 & 1 & 0 \\ 0 & 0 & 0 & 0 & 1 \\ 0 & 0 & 0 & 0 & 1 \end{bmatrix} \tag{48}$$

$$\boldsymbol{B} = \begin{bmatrix} 1 & 0 & 0 & 0 & 0 \\ 0.0444 & 0.9556 & 0 & 0 & 0 \\ 0 & 0.0741 & 0.9259 & 0 & 0 \\ 0 & 0 & 0.3571 & 0.6429 & 0 \\ 0 & 0 & 0 & 0.2222 & 0.7778 \end{bmatrix} \tag{49}$$

$$\boldsymbol{\Theta} = \{\theta_1 = [16; 0.4], \theta_2 = [90; 14.64], \theta_3 = [54; 10.80], \theta_4 = [14; 11.93], \theta_4 = [27; 8.20]\} \tag{50}$$

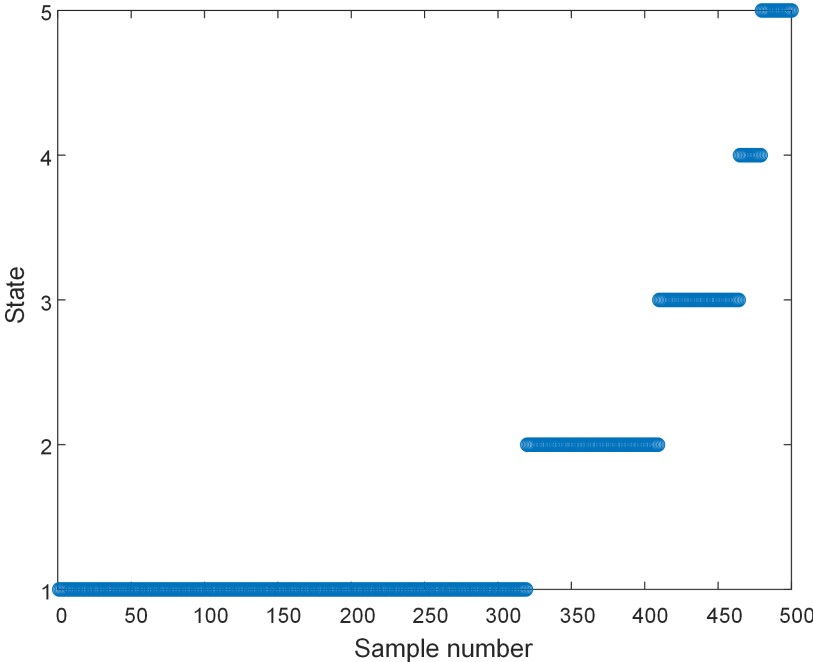

**Figure 8.** Observation sequence.

Six groups of sample data were selected. The state sequence recursive estimations that were obtained by the combination of SIR and HSMM are shown in Figure 9. Here, the blue solid lines represent the actual hidden state sequences, and the pink solid lines are the most likely hidden state sequences estimated by the combination of SIR and HSMM.

The recursive estimation error was then obtained, as shown in Figure 10. The estimation yielded with this method has a smaller error distribution. Although the estimation error may increase over time, it was still within an acceptable range. Hence, this method is able to estimate the sequence of hidden states through the recursion of the state at the previous moment and the observation at the current moment.

According to the estimation results of the current state, the RUL predictions of six groups of examples were further obtained. At the early stage under all six cases, the proposed method was able to estimate RUL values close to a constant at an early stage. Afterward, the estimates decreased almost linearly with time until the end of the available test samples, as shown in Figure 11. In addition, we also performed RUL prediction with conventional HSMM for comparative analysis, and the prediction results can also be seen in Figure 11. Evidently, the RUL values predicted by the method of this paper are closer to the actual value compared to the prediction results of the conventional HSMM.

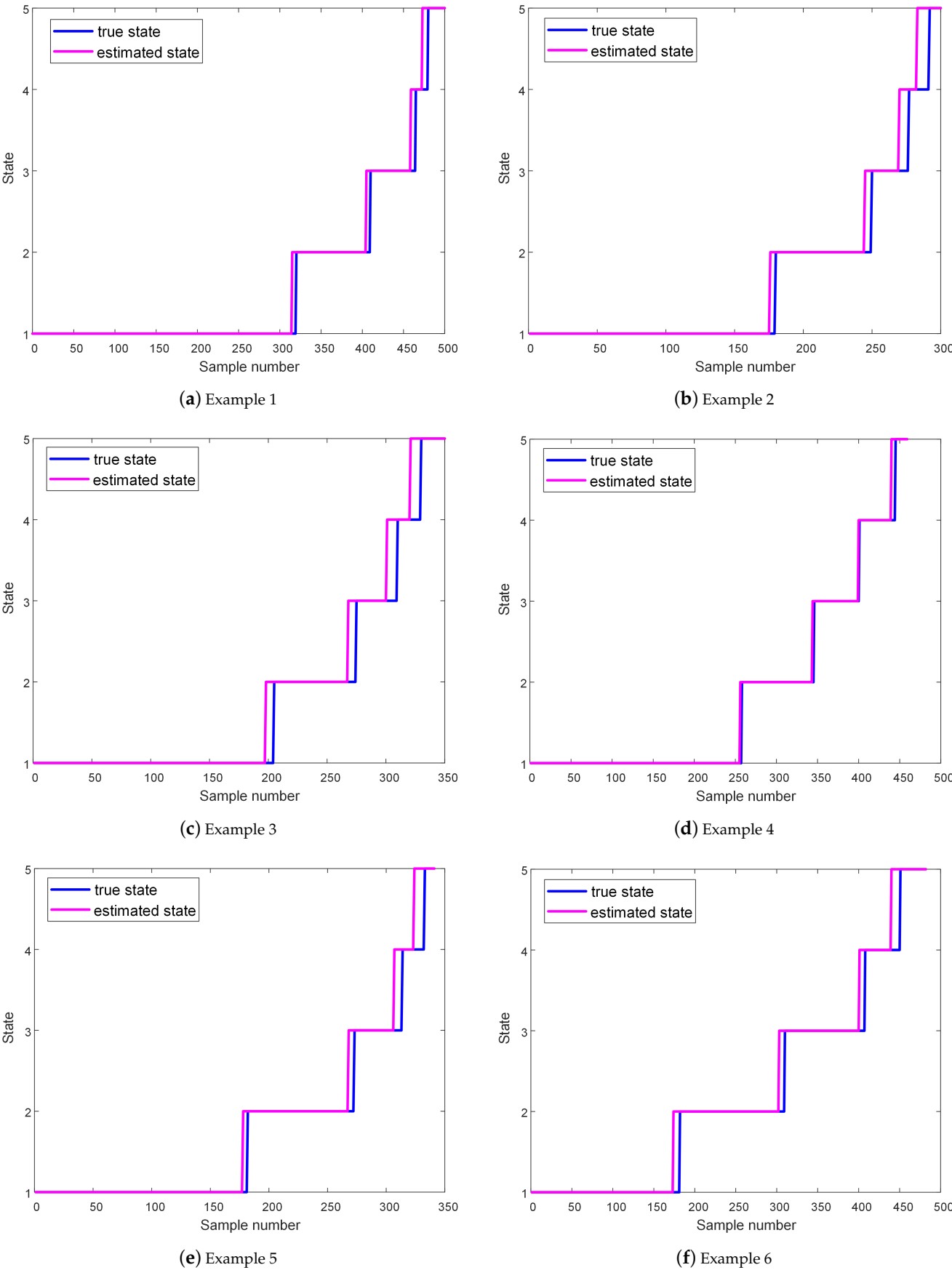

**Figure 9.** Health-state recursive-estimation results.

Two metrics were used to evaluate the RUL estimations: (1) Lifetime prediction performance (LPP), which can be measured by the percentage of RUL estimates within $+/-20\%$ of the true RUL [38] and (2) RMSE, which is widely used [39,40] to measure the accuracy of predictions and is the most commonly used regression metric [11]. Given the predicted RUL($RUL_m$) and true RUL($RUL_m^*$) under the time index $m$, the metric was then defined:

$$\text{RMSE} = \sqrt{\frac{1}{m}\sum_{t=1}^{m}(RUL_t - RUL_t^*)^2} \tag{51}$$

The results are shown in Table 3, which clearly illustrate that the proposed RUL prediction method was able to attain good prediction performance, achieving a very low RMSE. Moreover, the LPP reached over 90.00%, and the RMSE was noted to be lower than 4.9477. Each set of examples predicted better than the conventional HSMM. Thus, the RUL of PMSTM can be estimated accurately via combination of SIR and HSMM.

**Table 3.** Evaluation of RUL estimation results.

| Method | Index | 1 | 2 | 3 | 4 | 5 | 6 |
|---|---|---|---|---|---|---|---|
| HSMM+SIR | LPP | 95.16% | 93.33% | 90.00% | 96.95% | 95.16% | 96.48% |
| | RMSE | 3.2484 | 2.7805 | 4.9477 | 1.8166 | 4.9333 | 2.6907 |
| HSMM | LPP | 95.00% | 92.33% | 90.00% | 95.86% | 95.16% | 91.79% |
| | RMSE | 3.7523 | 3.7951 | 5.5617 | 2.2050 | 4.1889 | 3.1714 |

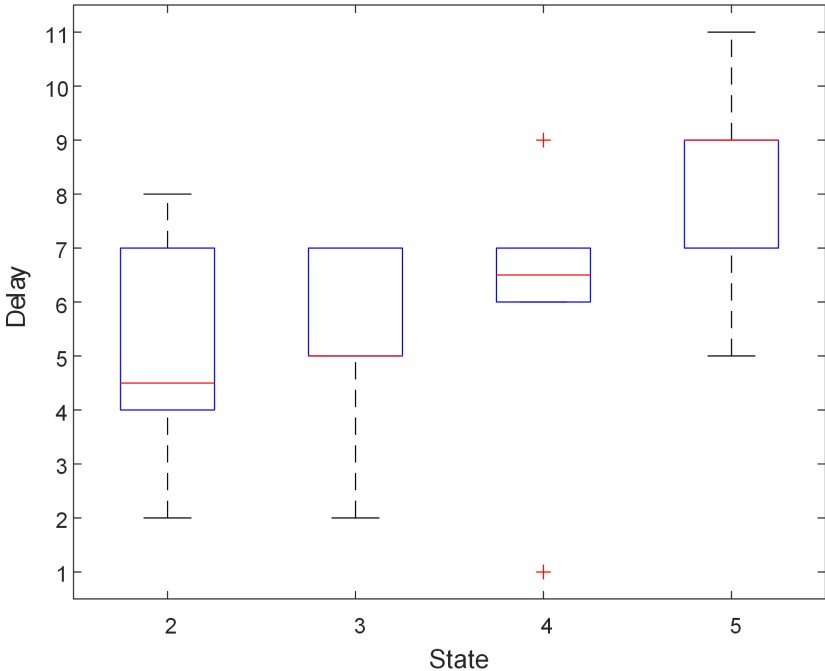

**Figure 10.** Health-state recursive-estimation error.

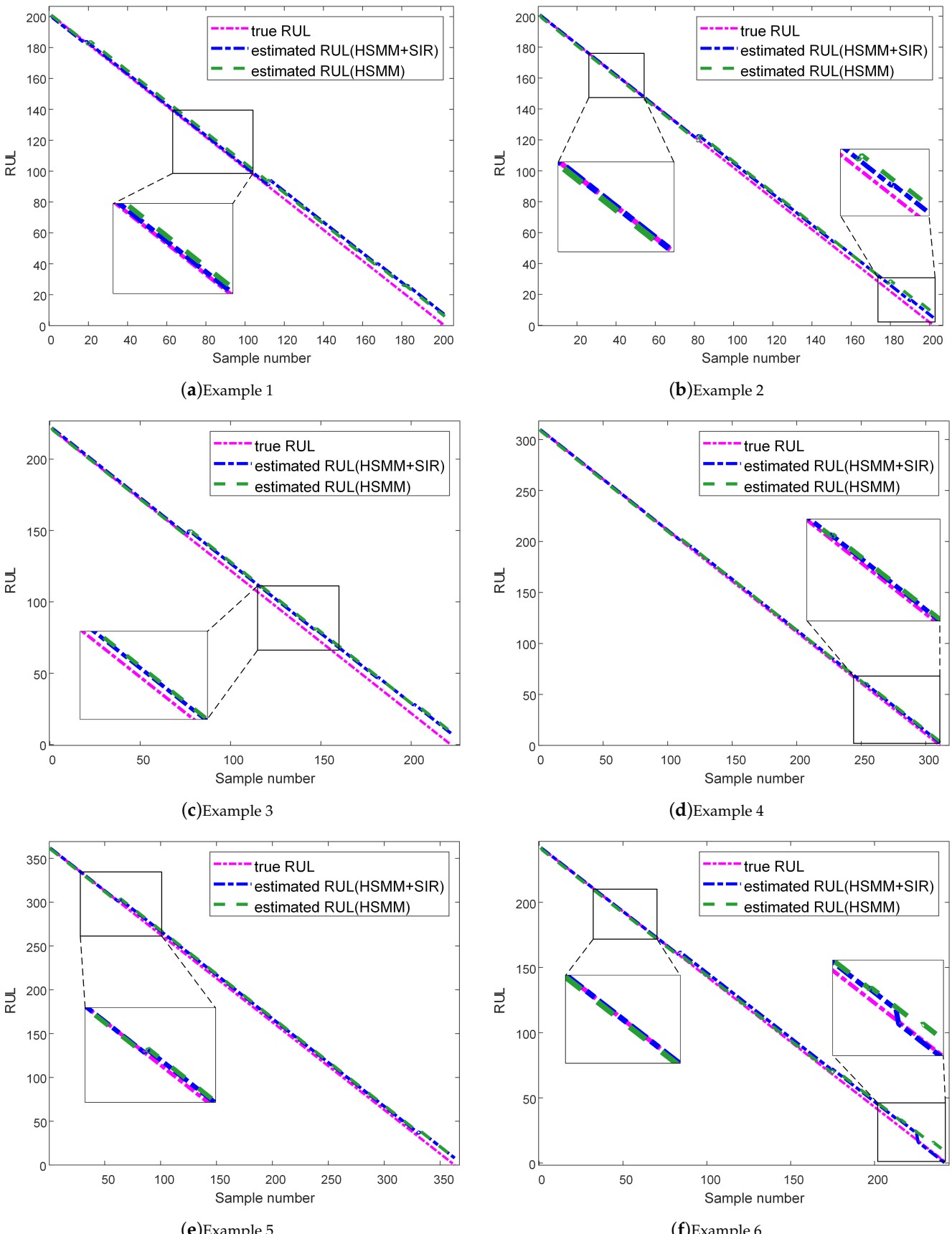

(**a**)Example 1

(**b**)Example 2

(**c**)Example 3

(**d**)Example 4

(**e**)Example 5

(**f**)Example 6

**Figure 11.** RUL estimation results.

## 6. Conclusions

This paper proposed a state estimation and RUL prediction method that combined SIR and HSMM with MFHI as the performance indicator for URT's PMSTMs. According to the proposed method, the signal is denoised by a wavelet, and the entropy weight method is then used to objectively determine the weights of multiple indicators in view of the amount of information in order to realize the prioritization of the state-characteristic parameters. The correlations between the selected feature parameters were reduced using PCA, and the MFHI was constructed through the corresponding weight assignment and fusion. HSMM training was then performed to describe the relationship between the internal state and observation sequence of the PMSTM, after which an overall performance degradation model was established. The observation sequence of the hidden state was observed to be directly recursive from the previous state through SIR, demonstrating the recursive optimal estimation of the internal state while realizing RUL estimation. By conducting simulation experiments, the state estimation and RUL prediction were performed by taking six groups of data as examples in order to verify the effectiveness of the algorithm. Additionally, compared with the conventional HSMM method, the results signify that the proposed method can generate more accurate predictions. In general, compared with other published methods, the method proposed in this paper was less computationally intensive and possessed more accurate computational results.

Future work might focus on finding more suitable methods to obtain observation sequences. Additionally, we will do some research on planning tests based on the proposed model.

**Author Contributions:** Conceptualization, G.T. and J.S.; methodology, G.T. and Y.Q.; software, G.T. and Y.Q.; validation, G.T. and S.W.; writing—original draft preparation, G.T.; writing—review and editing, S.W.; visualization, Y.Q.; supervision, S.W.; project administration, S.W. and J.S. All authors have read and agreed to the published version of the manuscript.

**Funding:** This research was funded by the National Natural Science Foundation of China (grants 51875014, 51620105010, 51575019).

**Institutional Review Board Statement:** Not applicable.

**Informed Consent Statement:** Not applicable.

**Data Availability Statement:** Not applicable.

**Acknowledgments:** The authors would like to thank all the teachers at the School of Automation Science and Electrical Engineering, Beijing University, for their support and advice in the completion of this work.

**Conflicts of Interest:** The authors declare no conflict of interest.

## Abbreviations

The following abbreviations are used in this manuscript:

| | |
|---|---|
| DCNN | Deep convolutional neural network |
| GMM | Gaussian mixture model |
| HI | Health index |
| HMM | Hidden Markov model |
| HSMM | Hidden Semi-Markov model |
| ISOMAP | Isometric mapping |
| LPP | Lifetime prediction performance |
| MFHI | Multi-parameter fusion health index |
| PCA | Principal component analysis |
| PMSTM | Permanent magnet synchronous traction motor |

| PSO | Particle swarm optimization |
| RMSE | Root mean square error |
| RNN | Recurrent neural network |
| RUL | Remaining useful life |
| SIR | Sample importance resampling |
| SNR | Signal-to-noise ratio |
| SVM | Support vector machine |
| URT | Urban rail transit |

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
