# Peer review of "State Estimation and Remaining Useful Life Prediction of PMSTM Based on a Combination of SIR and HSMM"

_sustainability, doi:10.3390/su142416810_

Round 1

Reviewer 1 Report

The authors presented the numerical method of analyses the data obtained from sensors in order to obtain a state estimation and estimate remaining useful life time for the Permanent magnet synchronous traction motors. The proposed method combined Hidden Semi-Markov model and Sampling importance resampling filter for estimation. Also method employs Wavelet denoising, Z-score standardization, Filter data with entropy weight method, Principal component analysis, Parameter fusion, Butterworth filter. At the end such processed data served for Hidden Semi-Markov method training and sent to Sample importance resampling.

They conducted simulation experiments, the state estimation and Remaining useful life prediction and they verified the effectiveness of the algorithm. They proved that the proposed method can generate more accurate predictions. It is emphasized that the proposed method is less computationally intensive and possessed more accurate computational compared with other published methods.

In my opinion the method of investigation is adequate, the presentation of results is clearly stated, conclusions are supported by results.

Author Response

Dear reviewer,

On behalf of my co-authors, we thank you very much for giving us an opportunity to revise our manuscript. We appreciate you very much for your constructive comments and suggestions on our manuscript entitled “State Estimation and Remaining Useful Life Prediction of PMSTM Based on Combination of SIR and HSMM” (ID: sustainability-1979765).

We have studied your comments carefully and have made revision. The major revised parts, especially the ones related to technical issues, are highlighted in green in the revised manuscript. Other minor changes, including order of presentation, spelling, and grammar are highlighted in blue. We have tried our best to revise our manuscript according to the comments from the reviewers. Please find the revised version, which we would like to submit for your kind consideration.

We would like to express our great appreciation to you and reviewers for comments on our paper. Looking forward to hearing from you.

Reviewer 2 Report

The article has great relevance in the academic field, however, in order to accept it, the reviewer's proposed corrections must be made.

1) I suggest that the evaluator revise the article in English;

 2) The author must rewrite the abstract according to a scientific article: purpose, introduction, methodology, results and conclusions;

3) The author of every scientific article uses acronyms such as PMSTM, SIR, HSMM, and MFHI. I suggest putting the meaning of those acronyms in the paper.

4) The intro does not describe the research problem. The author describes in the text only the advantages of the rail transport system, however, I did not observe the problem of the research. I propose in a paragraph outlining the problem of research,

5) The introduction has not rewritten the state of the art nor has it described the purpose of the search at the end;

6) I propose in the methodology to give a more detailed description of the method used and also to show why the author chose this method. Are there more advantages to this method than others? Describe. Does this method present disadvantages in comparison with other methods? Describe.

7) In the methodology, I suggest a more detailed explanation of the proposed methodology in Figure 1.

8) The methodology does not show Figure 2 underneath the paragraph. I also propose to explain the scheme of Figure 2.

9) The article contains relevant references, but also references to publications that are out of date. I suggest that references be included at least five years ago. Avoid referring to memoranda, theses and websites.

10) The submitter should explain the diagram in Figure 3.

11) The author did not distinguish the results from the discussion stage in the paper. I suggest you correct it in the paper.

12) The author should take a closer look at Figure 5. It would be interesting to compare with another writer with a citation.

13) It would be useful for the author to compare with another author with a quote from Table 3.   14) In conclusion, future work would be of interest.   15) Figures 9 (a) and 9 (b) do not have any X-axis legends.

Author Response

Dear reviewer,

On behalf of my co-authors, we thank you very much for giving us an opportunity to revise our manuscript. We appreciate you very much for your constructive comments and suggestions on our manuscript entitled “State Estimation and Remaining Useful Life Prediction of PMSTM Based on Combination of SIR and HSMM” (ID: sustainability-1979765).

We have studied your comments carefully and have made revision. The major revised parts, especially the ones related to technical issues, are highlighted in green in the revised manuscript. Other minor changes, including order of presentation, spelling, and grammar are highlighted in blue. We have tried our best to revise our manuscript according to the comments from the reviewers. Please find the revised version, which we would like to submit for your kind consideration.

We would like to express our great appreciation to you for comments on our paper. Looking forward to hearing from you.
